# Real-world evidence of the impact of obesity on residual teeth in the Japanese population: A cross-sectional study

Mayu Hayashi[1], Katsutaro Morino[2¤a]*, Kayo Harada[1], Itsuko Miyazawa[2¤b], Miki Ishikawa[1], Takako Yasuda[1], Yoshie Iwakuma[1], Kazushi Yamamoto[1], Motonobu Matsumoto[1], Hiroshi Maegawa[2], Atsushi Ishikado[1,2]*

1 Research and Development Department, Sunstar Inc., Takatsuki, Osaka, Japan, 2 Department of Medicine, Shiga University of Medical Science, Otsu, Shiga, Japan

¤a Current address: Institutional Research Office of Shiga University of Medical Science, Otsu, Shiga, Japan
¤b Current address: Education Center for Medicine and Nursing, Shiga University of Medical Science, Otsu, Shiga, Japan

* morino@belle.shiga-med.ac.jp (KM); atsushi.ishikado@jp.sunstar.com (AI)

**Data Availability Statement:** The data that support the findings of this study are available from MinaCare Co., Ltd., Japan. Restrictions apply to the availability of these data, which were used under

## Abstract

### Background

Tooth loss is associated with nutritional status and significantly affects quality of life, particularly in older individuals. To date, several studies reveal that a high BMI is associated with tooth loss. However, there is a lack of large-scale studies that examined the impact of obesity on residual teeth with respect to age and tooth positions.

### Objective

We assessed the impact of obesity on the number and position of residual teeth by age groups using large scale of Japanese database.

### Methods

This was a cross-sectional study of 706150 subjects that were included in the database that combined the data from health insurance claims and health check-up, those lacking information about BMI, HbA1c level, smoking status, and the number of residual teeth were excluded. Thus, a total of 233517 aged 20–74 years were included. Subjects were classified into 4 categories based on BMI, and the number of teeth was compared between age-groups. The percentage of subjects with residual teeth in each position was compared between groups with obesity (BMI $\geq$25.0 kg/m$^2$) and non-obesity. Logistic regression analysis was performed to clarify whether obesity predicts having <24 teeth.

### Results

Higher BMI was associated with fewer teeth over 40s (*P* for trend <0.0001 when <70s). Obesity was associated with the reduction of residual teeth in the maxillary; specifically, the molars were affected over the age 30. Smoking status further affected tooth loss at positions

license for this study. We did not have any privileged access to the data. Data are commercially available with MinaCare Co., Ltd., Japan. The dataset used was developed by MinaCare using anonymized data from dental claims and annual physical examinations of 2.8 million employees and their dependents in 2015. For further information, please email the representative at mc_info@minacare.co.jp.

**Funding:** This work was funded by Sunstar Inc.. This study was designed, conducted, and reported by the employees of the funder, Sunstar Inc., in collaboration with the investigators from Shiga University of Medical Science. Eight employees of the funder participated in the preparation, analysis, and interpretation of the data.

**Competing interests:** This work was funded by Sunstar Inc.. This study was designed, conducted, and reported by the employees of the funder, Sunstar Inc., in collaboration with the investigators from Shiga University of Medical Science. Eight employees of the funder participated in the preparation, analysis, and interpretation of the data. This does not alter our adherence to PLOS ONE policies on sharing data and materials.

that were not affected by obesity alone. After adjusting for age, sex, smoking status, and HbA1c ≥6.5%, obesity remained an independent predictive factor for having <24 teeth (ORs: 1.35, 95% CIs: 1.30–1.40).

## Conclusions

We found that an increase in BMI was associated with a decrease in the number of residual teeth from younger ages independently of smoking status and diabetes in the large scale of Japanese database.

## Introduction

Tooth loss has a significant impact on chewing and other systemic conditions such as digestion, speech, expression of emotions [1], and muscle strength [2]. This is particularly important in the elderly population, as tooth loss in this population is associated with nutritional status [3], healthy life expectancy [4], cognitive function [5], frailty [6], and mortality [5, 7]. Furthermore, tooth loss has a significant impact on the quality of life (QOL) of the elderly population [8]. Previous studies have suggested that a number of <24 teeth is predictor of subsequent edentulous jaw [9]. Periodontal disease and dental caries are two major factors that lead to tooth loss across age groups and races [10, 11]. Periodontal disease leads to inflammation of the gums and resorption of the alveolar bone due to the presence of bacteria inside plaques, and dental caries promote tooth resorption via root canal inflammation. Thus, Periodontal disease and dental caries can progress and eventually lead to tooth loss [12, 13]. Other factors that affect the number of residual teeth include age, smoking status, non-communicable diseases, such as diabetes, socioeconomic status, oral hygiene practices (e.g. frequency of tooth brushing and visits to dentists), and the oral health status, including the health of residual teeth [14–16].

Obesity is a precursor to diabetes, which is one of the known risk factors for tooth loss [17], and is characterized by excess fat accumulated on existing fat tissue. The World Health Organization (WHO) reported that the prevalence of obesity and excess weight is continuing to increase worldwide [18]. Obesity is largely associated with the systemic health of an individual, and is an important risk factor for metabolic syndrome, cardiovascular disease, kidney disease, and cancer [19, 20]. Obesity-induced chronic inflammation and dyslipidemia lead to insulin resistance, and increase the risk of periodontal disease development and progression [21]. They are also associated with the increase in the decayed, missing, and filled teeth (DMFT) index, which is an epidemiological index that describes the history of dental caries [22, 23]. Indeed, previous studies, including the Third National Health and Nutrition Examination Survey (NHANES III), revealed a correlation between a high BMI and the risk of periodontal disease [24, 25]. As for the relationship between obesity and the number of teeth, a high BMI is also associated with tooth loss [26–28]. However, there is a lack of large-scale studies that examined the impact of obesity on residual teeth with respect to age and tooth positions.

In the present study, we used a large database consisting of data from health insurance claims and health check-up of the Japanese population to assess the impact of obesity on the number and position of residual teeth by age groups. We further examined whether smoking status, which is a known risk factor for tooth loss, increases the impact of obesity on residual teeth. Lastly, we examined whether obesity predicts the risk of tooth loss independently from other risk factors, including diabetes and smoking status.

## Methods

### Study design and selection of study subjects

A retrospective, cross-sectional study using a commercially available Japanese healthcare database (MinaCare Co., Ltd., Japan) [15, 29–31] The database consisted of data from health insurance claims and health check-up between April 2015 and March 2016.

Subjects included in the database were individual workers and their family dependents aged between 6 and 89. In general, employment-based health insurance is only applied to those at large-scale retailers and manufacturers, and not to those who are self-employed or in primary industries. The data from health check-up included information on subject demographics, smoking status, vital signs, clinical laboratory test data, and management data. Among 706150 subjects that were included in the database that combined the data from health insurance claims and health check-up, those lacking information about BMI, HbA1c level, self-reported smoking status, and the number of residual teeth were excluded. Thus, a total of 233517 adults were included in the study (Fig 1).

### Variables

The analysis included demographics and clinical information collected from the electronic database. The number of residual teeth was calculated based on the dental receipt information of periodontitis, gingivitis, and chronic periodontitis, which was recorded in the health insurance claims. Thus, individuals with edentulous jaws were not included. BMI was calculated by dividing body weight in kilograms by the square of height in meters. In Japan, due to the fact that Asians are more prone to hypertension, dyslipidemia, and diabetes, BMI 18.5 or more and less than 25 is defined as "normal weight", 25.0 or more is defined as "obesity" [32]. Data on the levels of HbA1c (as defined by the National Glycohemoglobin Standardization Program (NGSP)), levels of fasting blood glucose, systolic and diastolic blood pressure, and smoking status were collected from the health check-up. Smoking status was self-reported, and was defined as smoking over 100 cigarettes on a regular basis during the past month and/or history of smoking for at least 6 months. The number of residual teeth and the percentage of subjects with residual teeth at each position were examined for 28 teeth, excluding the third molars. Based on the number of residual teeth, the percentage of subjects with residual teeth at each position was calculated as the proportion of all subjects with the respective residual teeth.

### Statistical analyses

Subjects were categorized according to their BMI ($<18.5$, 18.5–24.9, 25.0–29.9, and $\geq$30 kg/m$^2$) in each 10-year age groups. The median of each category was used to perform a linear trend test to assess the effects of age and obesity on the number of residual teeth. Furthermore, the chi-square test was performed to compare the percentage of subjects with residual teeth in each position between groups with obesity (BMI $\geq$25.0 kg/m$^2$) and non-obesity, as well as between subjects with obesity and non-obesity subjects with and without smoking status. Haberman's residual analysis was performed for variables that were identified as non-independent by the chi-square test. Four models were built to perform logistic regression analysis; specifically, age, sex, smoking status, and HbA1c $\geq$6.5% were included as independent variables by the forced entry method, and BMI $\geq$25.0 kg/m$^2$ by the step-wise method. Odds ratios with the 95% confidence intervals were calculated to clarify the predictor of tooth loss, which was defined as having $<24$ teeth.

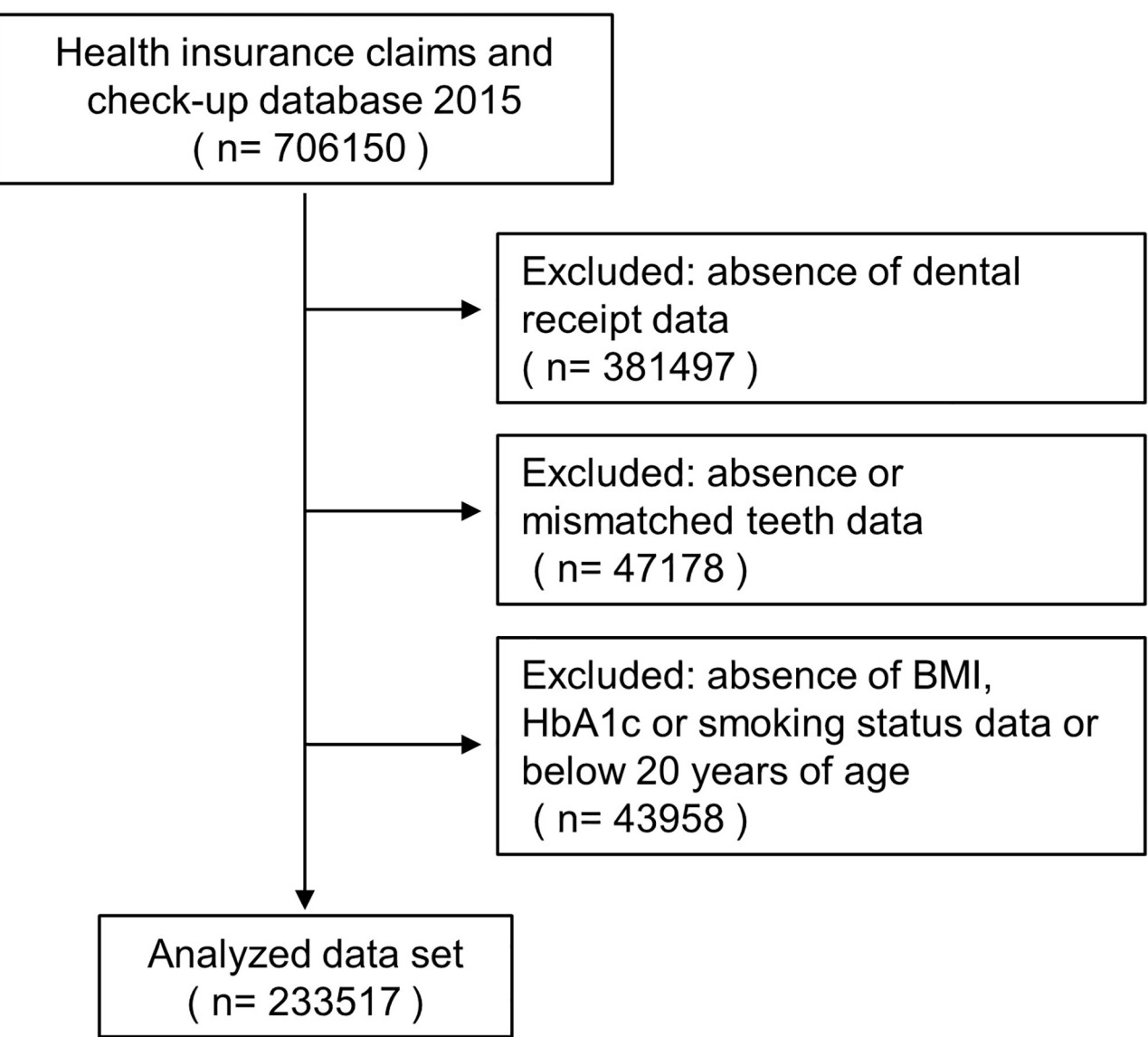

**Fig 1. Flowchart of the inclusion and exclusion of subjects in the health insurance claims and check-up database of 2015 provided by MinaCare Co., Ltd.**

All analyses were performed using SPSS (Statistical Package for Social Science) Statistics Ver. 26 (IBM, Inc., Armonk, NY, USA, Released on April 9, 2019), and *P*-values <0.05 were considered significant based on two-sided tests.

## Results

### Subject inclusion flowchart

The database of health insurance claims and health check-up included a total of 706150 subjects. Among them, those lacking information about BMI, HbA1c level, self-reported smoking status, and the number of residual teeth were excluded. Thus, a total of 233517 adults were included in the study (Fig 1).

**Table 1. Characteristics of the population from the medical database used in this study.** BMI; body mass index. Values are means (standard deviation) or percentage. Percentages of those missing data for fasting blood glucose and systolic/diastolic blood pressures were 10.1, <0.1, and <0.1% respectively.

| Age- groups / Characteristics | Total | 20–29 years | 30–39 years | 40–49 years | 50–59 years | 60–69 years | 70–74 years |
|---|---|---|---|---|---|---|---|
| | (n = 233517) | (n = 3958) | (n = 38805) | (n = 101171) | (n = 59113) | (n = 25408) | (n = 5062) |
| **Age (year)** | 47.7 (9.6) | 26.2 (2.4) | 36.1 (2.5) | 44.3 (2.8) | 53.8 (2.8) | 63.8 (2.7) | 71.9 (1.4) |
| **Sex** | | | | | | | |
| male | 110371 (47.3%) | 1716 (43.4%) | 15077 (38.9%) | 43626 (43.1%) | 31857 (53.9%) | 14642 (57.6%) | 3453 (68.2%) |
| female | 123146 (52.7%) | 2242 (56.6%) | 23728 (61.1%) | 57545 (56.9%) | 27256 (46.1%) | 10766 (42.4%) | 1609 (31.8%) |
| **BMI (kg/m2)** | 22.5 (3.6) | 20.9 (3.1) | 21.7 (3.6) | 22.5 (3.8) | 22.9 (3.6) | 22.8 (3.1) | 22.6 (2.8) |
| **Fasting plasma glucose (mg/dL)** | 93.8 (15.9) | 85.3 (10.8) | 88.5 (10.8) | 92.0 (14.5) | 97.0 (17.7) | 100.6 (18.5) | 102.8 (17.2) |
| **HbA1c (%)** | 5.5 (0.5) | 5.2 (0.3) | 5.3 (0.4) | 5.4 (0.5) | 5.6 (0.6) | 5.7 (0.6) | 5.8 (0.6) |
| **Systolic blood pressure (mmHg)** | 116.1 (16.2) | 109.8 (12.4) | 109.2 (13.5) | 113.9 (15.2) | 119.6 (16.3) | 125.2 (16.6) | 128.6 (16.5) |
| **Diastolic blood pressure (mmHg)** | 72.5 (11.7) | 65.7 (8.8) | 67.8 (10.3) | 71.6 (11.6) | 75.7 (11.8) | 76.7 (10.9) | 75.8 (10.4) |
| **Current smoker** | 43273 (18.5%) | 684 (17.3%) | 6884 (17.7%) | 20005 (19.8%) | 11817 (20.0%) | 3513 (13.8%) | 370 (7.3%) |
| **Number of residual teeth** | 26.3 (3.2) | 27.6 (1.3) | 27.4 (1.5) | 27.0 (2.0) | 25.8 (3.4) | 23.8 (5.1) | 22.3 (6.1) |
| **Number with < 24 teeth** | 24110 (10.3%) | 20 (0.5%) | 603 (1.6%) | 4512 (4.5%) | 8622 (14.6%) | 8198 (32.3%) | 2155 (42.6%) |

## Characteristics of study subjects

The characteristics of the entire study cohort and subjects by age groups are presented in Table 1. The following variables were examined: number of subjects, age, sex, BMI, fasting blood glucose level, HbA1c level, systolic blood pressure, diastolic blood pressure, smoking status, the total number of residual teeth (excluding the third molars), and the percentage of subjects with having <24 teeth. Data on fasting blood glucose, systolic blood pressure, and diastolic blood pressure were missing for 10.1, <0.1, and <0.1% of all study subjects, respectively.

The mean age of the study subjects was 47.7 ± 9.6 years. Although the proportion of male subjects was relatively small in the entire cohort (47.3%), male subjects were the majority in age groups over 50 years. The mean BMI in the entire cohort was 22.5 ± 3.6 kg/m². Smoking status was reported in 18.5% of the entire cohort, and was most common for subjects in their 40s (19.8%) and 50s (20.0%). The number of residual teeth decreased with age, with a mean of 27.6 ± 1.3, 27.4 ± 1.5, 27.0 ± 2.0, 25.8 ± 3.4, 23.8 ± 5.1, and 22.3 ± 6.1 in subjects in their 20s, 30s, 40s, 50s, 60s, and 70s, respectively (Table 1).

## Association between BMI and the number of residual teeth by age groups

The number of residual teeth decreased with age (Fig 2). As the BMI increased, the number of residual teeth also increased in subjects in their 20s (*P* for trend = 0.0013) and 30s (*P* for trend <0.0001), whereas it decreased in subjects in their 40s-60s (*P* for trend <0.0001 and in their 70s (*P* for trend = 0.0001).

The number of residual teeth in subjects in their 50s was 26.0 ± 3.2 for those with a low BMI (<18.5 kg/m²), whereas it was 25.2 ± 3.9 in those with a high BMI (≥30.0 kg/m²). This represented a difference of 0.8. Similarly, the difference in the number of residual teeth between the low and high BMI groups in subjects in their 60s was 0.7 (24.0 ± 5.0 and 23.3 ± 5.1, respectively). For subjects in their 50s, 60s, and 70s, the differences in the number of residual teeth with respect to the lowest BMI group (<18.5 kg/m²) were 0.1, 0.2, and 0.2 in those with a BMI between 18.5–24.9 kg/m², 0.6, 0.7, and 0.9 in those with a BMI between 25.0–29.9 kg/m², and 0.8, 0.8, and 2.4 in those with a BMI ≥30.0 kg/m², respectively. Thus, the

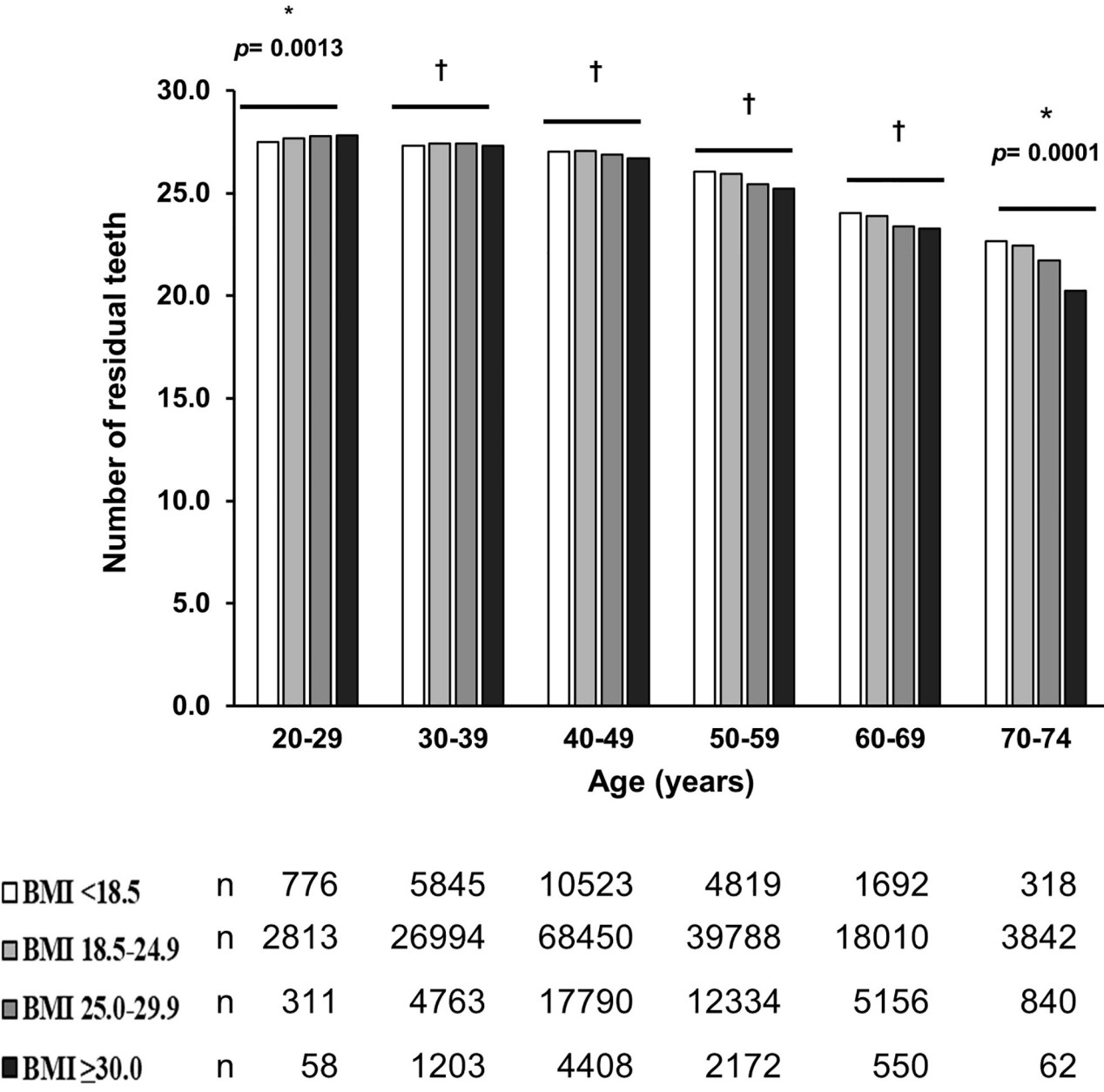

**Fig 2. Number of residual teeth by BMI class in each 10-year age group.** The values are the mean and sample sizes in each age- and BMI- category. Significant linear trend across BMI classes, *; $p < 0.05$, †; $p < 0.0001$.

higher the BMI is, the more significant the difference in the number of residual teeth especially, in case of those with a BMI $\geq 25.0$ kg/m² (Fig 2).

## Percentage of subjects with residual teeth at each position

The comparison of the percentage of subjects (ages 30–69) with residual teeth at each position between groups with non-obesity (BMI $<25.0$ kg/m²) and obesity (BMI $\geq 25.0$ kg/m²) is shown in Fig 3 and S1 Movie. The comparison was made within the same age group, and positions with a significant reduction in the percentage of subjects with residual teeth are indicated

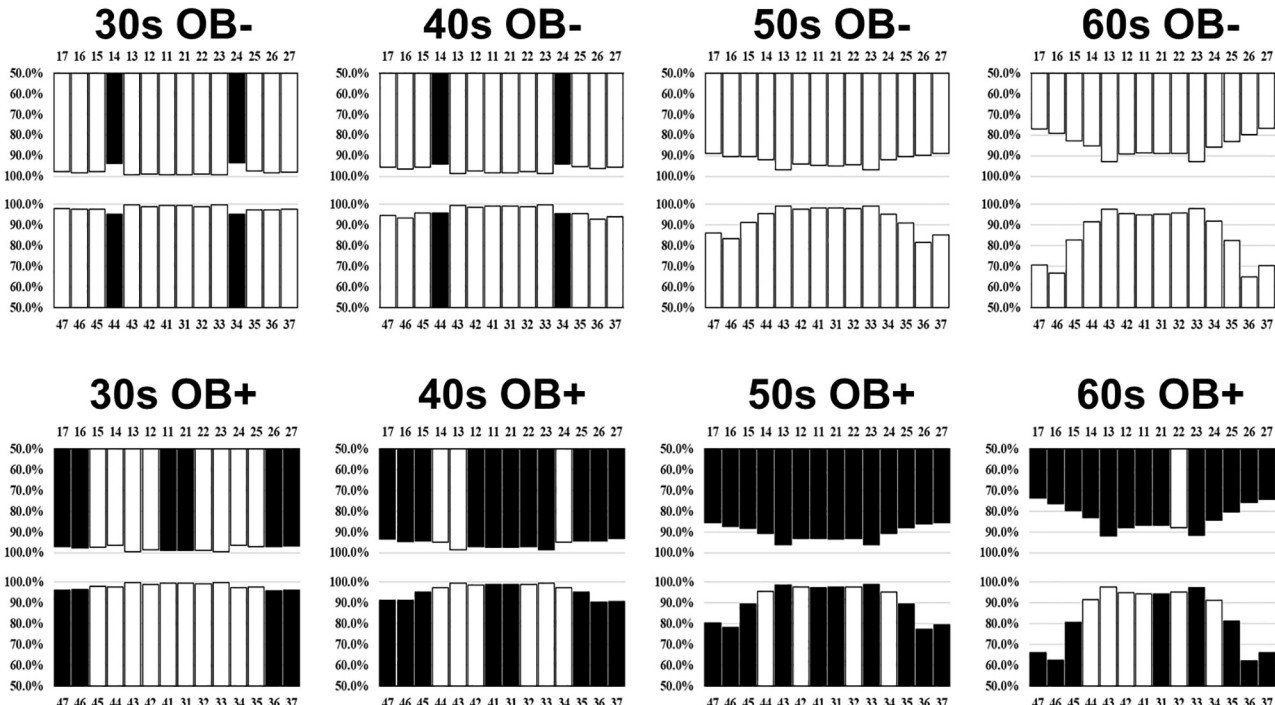

**Fig 3. Percentage of subjects with residual teeth at each position in groups with non-obesity (BMI <25.0 kg/m²) and obesity (≥25.0 kg/m²) by age groups (30s-60s).** The percentage of subjects with residual teeth was calculated as the proportion of subjects that have a residual tooth at the particular position. The percentage of subjects with residual teeth was compared between the groups with obesity and non-obesity in the same age group, and positions with a significantly lower percentage of subjects having residual teeth are shown in black (*p* <0.05). OB+: obesity, OB-: non-obesity.

in black. Compared to the group with non-obesity, the percentage of subjects with obesity with residual teeth was significantly lower at 10 positions (6 maxillary, 4 mandibular) for those in their 30s, 19 positions (11 maxillary, 8 mandibular) for those in their 40s, 24 positions (14 maxillary, 10 mandibular) for those in their 50s, and 21 positions (13 maxillary, 8 mandibular) for those in their 60s. In contrast, there were 4 positions (2 maxillary, 2 mandibular) for which the percentage of subjects with residual teeth was significantly higher among subjects with obesity aged between 30s and 40s. A significant percentage of subjects with obesity aged between 30s-60s had lost the molar and maxillary central incisor, whereas a significant percentage of those aged between 30s and 40s had residual first premolars.

Residual tooth loss was the most common at the maxillary left first premolar (93.5%) in subjects with non-obesity in their 30s. In contrast, it was the most common at the mandibular left first molar in subjects with obesity in their 30s and in their 40s-60s; specifically, the percentage of those with a residual mandibular left first molar was 95.8% among subjects with obesity in their 30s, 92.8 and 90.3% among non-obesity and obesity in their 40s, 81.6 and 77.4% among non-obesity and obesity in their 50s, and 65.0 and 62.3% among non-obesity and obesity in their 60s. The impact of obesity on tooth loss was the most notable on the mandibular right second molar across all age groups. Between the groups with non-obesity and obesity, the differences in the percentage of subjects with a residual mandibular right second molar were 1.8, 3.5, 5.6, and 4.6% in subjects in their 30s, 40s, 50s, and 60s, respectively.

The comparison of the percentage of subjects with residual teeth at each position between subjects with non-obesity and obesity with and without smoking status is shown in Fig 4. Four groups in each age group were compared, and positions with a significantly lower percentage of subjects having a residual tooth than the expected value are shown in black. Among groups

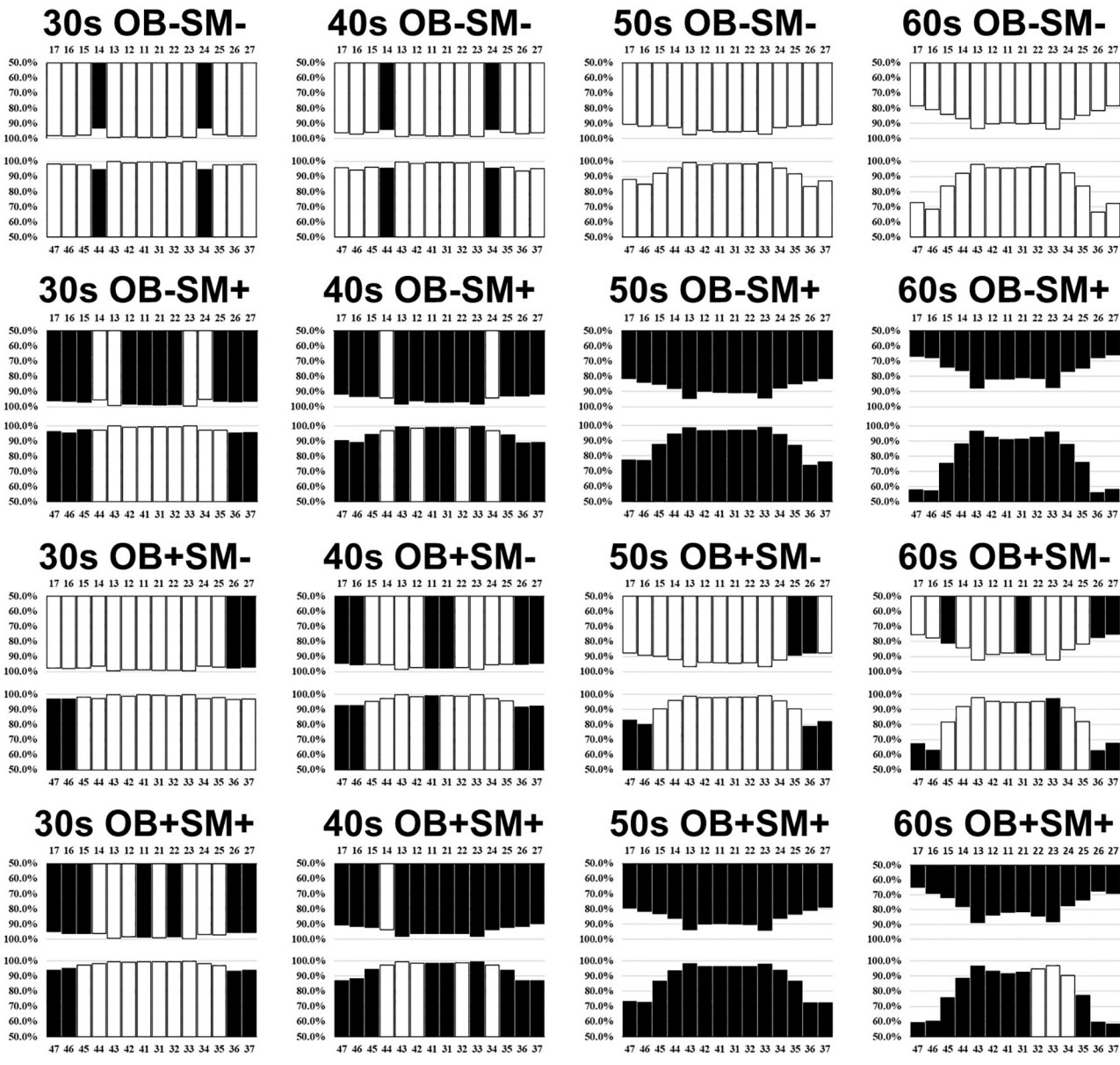

**Fig 4. Percentage of obesity/ non-obesity/ smoking/ non-smoking subjects with residual teeth at each position by age groups (30s-60s).** Percentage of subjects with residual teeth was calculated as the proportion of subjects that have a residual tooth at the particular position. The percentage of subjects with residual teeth was compared among the 4 groups in the same age group, and positions with a significantly lower percentage of subjects having a residual tooth than the expected value are shown in black ($p < 0.05$). OB-; non-obesity, OB+; obesity SM-; non-smoking, SM+; smoking.

with non-obesity non-smoking (OB-SM-), non-obesity smoking (OB-SM+), obesity non-smoking (OB+SM-), and obesity smoking (OB+SM+), the OB-SM+ group had the highest number of positions with a significantly lower percentage of subjects having residual teeth than the expected value. These included 15 positions (10 maxillary, 5 mandibular) in those in their 30s and 22 positions (12 maxillary, 10 mandibular) in those in their 40s. Similarly, all tooth positions were significantly affected in OB-SM+ subjects in their 50s and 60s. Loss of residual teeth was also common in the OB+SM+ group, affecting 11 positions (7 maxillary, 4 mandibular), 22 positions (13 maxillary, 9 mandibular), all, and 25 positions (14 maxillary, 11

mandibular) in subjects in their 30s, 40s, 50s, and 60s, respectively. On the other hand, the number of positions with a significantly higher percentage of subjects having residual teeth than the expected value was the highest in the OB-SM- group, with 13 positions (9 maxillary, 4 mandibular) in those in their 30s, 21 positions (12 maxillary, 9 mandibular) in those in their 40s, and all positions in those aged 50s and 60s.

In the OB+SM+ group, positions at which the percentage of subjects having residual teeth was significantly lower than the expected value included the molar, maxillary right second premolar, maxillary right central incisor, and the maxillary left lateral incisor in those in their 30s-60s and mandibular molar, mandibular second premolar, mandibular central incisor, an all maxillary positions in those in their 40s-60s (except the maxillary right first premolar in their 40s). The percentage of subjects having residual first premolars was significantly higher than the expected value except OB-SM- subjects in their 30s and 40s (except the maxillary first premolars in OB-SM+ and OB+SM+ subjects in their 40s).

Positions with the lowest percentage of subjects having residual teeth included the maxillary left first premolar in OB-SM- subjects in their 30s (93.2%), mandibular left second molar in OB+SM+ subjects in their 40s (86.9%) and 50s (72.3%), and mandibular left first molar in OB-SM+ subjects in their 60s (55.9%). The impact of obesity and smoking on tooth loss was most notable on the mandibular left first molar in those in their 30s, mandibular right second molar in those in their 40s, mandibular left second molar in those in their 50s, and maxillary left first molar in those in their 60s. Compared with the OB-SM- group, the differences in the percentage of subjects with residual teeth at these positions were 4.5, 8.7, 14.9, and 14.1%, respectively.

Further examination for the impact of smoking on the number of residual teeth in subjects with obesity demonstrated that OB+SM+ subjects in their 30s, 40s, 50s, and 60s had 2.8-, 2.0-, 4.7-, 2.8- times more number of positions than OB+SM- subjects at which the number of residual teeth was significantly lower than the expected value. In the OB+SM- group, the percentage of subjects with residual teeth was significantly lower than the expected values at the maxillary left first molar, maxillary left second molar (except those in their 50s), mandibular right molar, and mandibular left molar (except those in their 30s). In addition to these positions, OB+SM+ subjects had a significantly lower number of residual teeth than the expected values at the canine tooth (except those in their 30s), right molar (except those in their 40s), right second premolar (except those in their 60s), right lateral incisor (except those in their 30s), right central incisor (except those in their 40s), left first premolar (except those in their 30s), and left lateral incisor of the maxilla in addition to the second premolar (except those in their 30s) and left central incisor (except those in their 30s) of the mandible.

### Risk factors for tooth loss (<24 teeth)

Having <24 teeth, is a risk factor for edentulous jaw [9]. In the Healthy Japan 21, the Ministry of Health, Labour and Welfare proposed specifically targeting preservation of over 24 teeth by the age of 60 in order to achieve 20 teeth at the age of 80. We performed a logistic regression analysis using 4 models adjusted for other risk factors (Model 1: BMI $\geq$25.0 kg/m$^2$, Model 2: Model 1 + sex + age, Model 3: Model 2 + smoking status, Model 4: Model 3 + HbA1c $\geq$6.5%) to examine whether obesity was an independent risk factor for the tooth loss (having <24 teeth). In each model, the ORs for a BMI $\geq$25.0 kg/m$^2$ was 1.47 (95%CIs: 1.43–1.52), 1.39 (95% CIs: 1.35–1.44), 1.39 (95%CIs: 1.34–1.44), and 1.35 (95%CIs: 1.30–1.40), respectively (Table 2).

## Discussion/conclusion

Our study led to two novel findings. First, we demonstrated that the increase in BMI is associated with a decrease in the number of residual teeth from younger age. Second, we showed

**Table 2. The ORs for fewer than 24 residual teeth.**

|  | Model; odds ratios (95% CIs) | | | |
|  | 1 | 2 | 3 | 4 |
|---|---|---|---|---|
| BMI $\geq$25 kg/m$^2$ | 1.47 (1.43–1.52) | 1.39 (1.35–1.44) | 1.39 (1.34–1.44) | 1.35 (1.30–1.40) |
| sex |  | 0.85 (0.82–0.87) | 1.03 (1.00–1.07) | 1.04 (1.01–1.08) |
| age |  | 1.12 (1.12–1.12) | 1.13 (1.13–1.13) | 1.13 (1.12–1.13) |
| smoking |  |  | 2.46 (2.37–2.55) | 2.44 (2.36–2.53) |
| HbA1c $\geq$6.5% |  |  |  | 1.37 (1.30–1.46) |

Model 1; BMI $\geq$25.0 kg/m2

Model 2; Model1+sex, age

Model 3; Model2+smoking

Model 4; Model3+HbA1c $\geq$6.5%

that obesity is associated with the loss of residual molars in subjects over the age 30, and that smoking status further affected tooth loss at positions that were not affected by obesity alone.

The number of residual teeth decreased with increasing BMI in subjects over the age of 40 (Fig 2). This is consistent with previous studies demonstrated that a high BMI and energy intake are associated with a reduced number of residual teeth in women aged between 37 and 60 [27], and that BMI ($\geq$30 kg/m$^2$) and abdominal obesity are associated with tooth loss in individuals younger than 60 years of age, regardless of their age or sex [28]. Our study provides additional insight because there are no studies to date that used large-scale datasets to examine the association between BMI and the number of residual teeth by age groups. Previous studies reported that high BMI is associated with the progression of periodontal disease [24, 25]. Furthermore, the intake of sugar and sweetened beverages is positively correlated with both weight gain and the DMFT index, which is an epidemiological index that describes the history of dental caries [23, 33]. Collectively, these studies suggested that the progression of periodontal disease and dental caries could reduce the number of residual teeth in the populations with obesity. Moreover, the lower class of residual teeth number was associated with more deteriorated clinical parameters of lifestyle diseases including diabetes (fasting plasma glucose $\geq$126 mg/dL and HbA1c $\geq$6.5%), hypertension (systolic blood pressure $\geq$140 mmHg and/or diastolic blood pressure $\geq$90 mmHg) or hyperlipidemia (triglyceride $\geq$150 mg/dL or low density lipoprotein $\geq$140 mg/dL or high density lipoprotein <40 mg/dL) (S1 Fig) and the consequent higher applicable number of each disease (S2 Fig), suggesting plausible involvement of increased risk in teeth loss by obesity in the higher incidence of lifestyle diseases.

On the other hand, we demonstrated that a high BMI is associated with a greater number of residual teeth in subjects in their 20s and 30s (Fig 2). Furthermore, the percentage of subjects with obesity in their 30s and 40s who had their first premolars was higher than that expected values (except for maxillary molars in OB+SM+ subjects in their 40s) (Figs 3 and 4). Tooth extraction due to orthodontic treatment is common in individuals between the ages of 10s-30s worldwide [10, 11]. In the Japanese population, extraction of the first premolars as a result of orthodontic treatment is more common than for other teeth [34]. Moreover, people with obesity often have poorer oral hygiene practices [35]. Thus, young subjects with obesity may have fewer tooth extractions as a result of orthodontic treatments due to poorer awareness of dental maintenance. Therefore, the low prevalence of tooth extraction as a result of orthodontic treatment in subjects with obesity in their 30s and 40s may explain the association between BMI and the number of residual teeth we observed in this population.

The percentage of subjects with residual teeth, especially the molars, decreased with age (Fig 3). This finding is consistent with previous studies demonstrating that the risk of losing the molars increases with age [36, 37], similar to the percentage of subjects with residual teeth by position and age reported in the 2016 Survey of Dental Diseases [38]. In subjects with obesity, significant tooth loss was more common in the maxillary than in the mandibular area; specifically, the percentage of subjects with residual molars decreased in subjects over the age of 30 (Fig 3). Periodontal disease and dental caries may have affected particularly the loss of the molars in subjects with obesity. Periodontal disease leads to secondary occlusal trauma and increases the risk of tooth loss. The first molars have the greatest relative occlusal force, followed by the second molars [39, 40]. As people with obesity have a higher risk of developing progressive periodontal disease [24, 25], occlusal trauma may have significantly impacted the molars. Furthermore, tooth extraction due to dental caries is most common for the molars [38]. As people with obesity have a high DMFT index [23], dental caries likely impacted the loss of the molars in our subjects with obesity. Indeed, a previous study demonstrated that poor occlusion and the use of incompatible prostheses especially in the molars are associated with malnutrition [41]. Many studies also found that prosthetics alone are not sufficient for improving the nutritional status [42–44]. Collectively, these studies suggested that the loss of natural teeth, especially the molars, in people with obesity may negatively affect the nutritional status and promote obesity.

We further added smoking status to our analysis, and found that the number of positions with a significantly lower percentage of subjects having residual teeth than the expected value was markedly greater in the OB-SM+ and OB+SM+ groups than in the OB-SM- and OB+SM- groups (Fig 4). This is consistent with previous studies reporting that smoking is associated with periodontal status and future tooth loss [45]. Furthermore, compared with subjects in the OB+SM- group, subjects in the OB+SM+ group had a greater number of positions at which a significantly lower percentage of subjects had residual teeth than the expected value. Smoking further affected positions that were not affected by obesity alone, such as the maxillary left lateral incisors; these included the 7, 11, 22, and 17 positions in subjects in their 30s, 40s, 50s, and 60s, respectively (Fig 4). Therefore, we confirmed that smoking status increases the risk of tooth loss in people with obesity, including at positions that are not affected by obesity alone. A recent systematic review demonstrated that individuals who quit smoking did not have an increased risk of tooth loss compared with individuals with no smoking history [46]. This suggests that smoking cessation is important for preventing tooth loss in people with obesity.

We performed a logistic regression analysis and demonstrated that obesity remained a risk factor for tooth loss (having <24 teeth) independently from other risk factors such as smoking status and diabetes (Table 2). This is consistent with previous studies demonstrating an association between a high BMI and tooth loss [27, 28]. Obesity is a precursor to diabetes, a known risk factor for tooth loss, and is strongly associated with the development of type 2 diabetes [47]. Thus, the risk of tooth loss due to diabetes is probably increasing from its preclinical stage of obesity and abnormal glucose tolerance.

Our study has several advantages over previous studies. They include the large sample size, and the use of accurate and large-scale medical information that enabled multiple sub-group analysis based on age, sex, BMI, and smoking status. Furthermore, there was little impact by selection bias as our data were obtained from medical insurance associations. The major limitation of the study was that the study cohort comprised only those individuals who visited dental clinics and were diagnosed with periodontal and other dental diseases, and it did not include those with edentulous jaw. However, as the management and treatment of periodontal disease involve the entire oral cavity, we believe that the dental formula of the dental insurance claims is an appropriate estimate of the number of teeth. Another limitation was the cross-

sectional study design, which is not sufficient to confirm a causal relationship between obesity and the number of residual teeth. Further studies are needed to validate our findings.

In conclusion, we found that 1) an increase in BMI was associated with a decrease in the number of residual teeth from younger age, 2), obesity was associated with the loss of residual molars in subjects over the age 30, and 3) smoking status has an additional impact on tooth loss at positions that were different from those that were impacted by obesity. We also demonstrated that obesity predicted the risk of tooth loss (having <24 teeth) independently from smoking status and diabetes. In addition to early prevention of periodontal disease and dental caries, which directly cause tooth loss, our study suggests that weight loss and smoking cessation also becomes more essential to prevent tooth loss in people with obesity. A recent study suggested that frequent tooth brushing and the fewer number of missing teeth reduce the risk of developing new-onset diabetes [48]. In people with obesity, the risk of diabetes may also be reduced if they keep practicing oral care from an early age and maintain as many natural teeth as possible.

## Supporting information

**S1 Fig. Clinical parameters of lifestyle disease by the class of residual teeth number in each 10-year age group.** Significant linear trend across classes of residual teeth number, *; $p$ <0.05, †; $p$ <0.0001.
(PDF)

**S2 Fig. Applicable number for lifestyle diseases by the class of residual teeth number in each 10-year age group.** The values are the mean and sample sizes in each age—and residual teeth number- category. Lifestyle diseases refer to diabetes (FPG ≥126 mg/dL and HbA1c ≥6.5%), hypertension (SBP ≥140 mmHg and/or DBP ≥90 mmHg) and hyperlipidemia (TG ≥150 mg/dL or LDL ≥140 mg/dL or HDL <40 mg/dL). Significant linear trend across classes of residual teeth number, †; $p$ <0.0001.
(PDF)

**S1 Movie. Percentage of subjects with residual teeth at each position in groups with non-obesity (BMI <25.0 kg/m$^2$) and obesity (≥25.0 kg/m$^2$) by age groups (30s-60s).** The percentage of subjects with residual teeth was calculated as the proportion of subjects that have a residual tooth at the particular position. OB+: obesity, OB-: non-obesity.
(MP4)

## Acknowledgments

The authors thank Dr. Satoshi Shizukuishi, Dr. Midori Tsuneishi and Mr. Takeshi Kanata for their expert technical assistance.

## Author Contributions

**Conceptualization:** Mayu Hayashi, Katsutaro Morino, Kayo Harada, Itsuko Miyazawa, Atsushi Ishikado.

**Data curation:** Kayo Harada, Miki Ishikawa, Takako Yasuda.

**Formal analysis:** Mayu Hayashi, Katsutaro Morino, Kayo Harada, Itsuko Miyazawa.

**Supervision:** Katsutaro Morino, Atsushi Ishikado.

**Validation:** Mayu Hayashi, Yoshie Iwakuma, Atsushi Ishikado.

**Visualization:** Yoshie Iwakuma.

**Writing – original draft:** Mayu Hayashi, Kayo Harada.

**Writing – review & editing:** Mayu Hayashi, Katsutaro Morino, Kayo Harada, Itsuko Miyazawa, Miki Ishikawa, Takako Yasuda, Yoshie Iwakuma, Kazushi Yamamoto, Motonobu Matsumoto, Hiroshi Maegawa, Atsushi Ishikado.

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
