## [Decision Letter · Decision Letter 0]

18 Mar 2022

PONE-D-21-34039Real-world evidence of the impact of obesity on residual teeth in the Japanese population: A cross-sectional studyPLOS ONE

Dear Dr. Morino,

Thank you for submitting your manuscript to PLOS ONE. After careful consideration, we feel that it has merit but does not fully meet PLOS ONE’s publication criteria as it currently stands. Therefore, we invite you to submit a revised version of the manuscript that addresses the points raised during the review process. Please ensure that your decision is justified on PLOS ONE’s publication criteria and not, for example, on novelty or perceived impact.

We look forward to receiving your revised manuscript.

Kind regards,

Rohit Kunnath Menon

Academic Editor

PLOS ONE

Journal Requirements:

Reviewers' comments:

Reviewer's Responses to Questions

**Comments to the Author**

1. Is the manuscript technically sound, and do the data support the conclusions?

Reviewer #1: Yes

Reviewer #2: Yes

2. Has the statistical analysis been performed appropriately and rigorously? 

Reviewer #1: Yes

Reviewer #2: Yes

3. Have the authors made all data underlying the findings in their manuscript fully available?

Reviewer #1: Yes

Reviewer #2: Yes

4. Is the manuscript presented in an intelligible fashion and written in standard English?

Reviewer #1: Yes

Reviewer #2: Yes

5. Review Comments to the Author

Reviewer #1: This manuscript if of intertest giving the ongoing debate as to the relationship between nutritional status/ anthropometrics and oral health. The study benefits from its real-world data course and large sample size.

Abstract: The abstract is clear and provides a useful summary of the study. However, the convulsion should be based on study findings rather than proposed future work (as this has not been evaluated).

Introduction: A brief succinct introduction is provided with a clear rationale for the study.

Methods: The study benefit from a large real-world sample of over 700,000 subjects (of which >230,000 were included in this study). In addition, a large number of variables were considered from biological to self-ratings. The statistical approach is clearly outlined and appropriate.

Results: A detailed results section is provided and supported by figures and tables. In additional supplementary files are available providing additional insight.

Discussion: A well written discussion is provided with interpretation of the findings compared with the literature. Some novel theories and aspects are provided. The conclusion is adequate and this should eb used to inform the abstract’s conclusion.

Reviewer #2: Few suggestions to be included

In statistical analysis, it is better to include full name of SPSS along with release date and version.

According to WHO, BMI >30 is considered as obese and BMI >25 and <30 is considered as overweight. In your statistical analysis, Obesity has been considered as BMI ≥25. Please justify.

6. PLOS authors have the option to publish the peer review history of their article (what does this mean?). If published, this will include your full peer review and any attached files.

Reviewer #1: No

Reviewer #2: **Yes: **Shilpa Gunjal

---

## [Author Response · Author response to Decision Letter 0]

6 Apr 2022

Reviewer #1

This manuscript if of intertest giving the ongoing debate as to the relationship between nutritional status/ anthropometrics and oral health. The study benefits from its real-world data course and large sample size.

Abstract: The abstract is clear and provides a useful summary of the study. However, the convulsion should be based on study findings rather than proposed future work (as this has not been evaluated).

Introduction: A brief succinct introduction is provided with a clear rationale for the study.

Methods: The study benefit from a large real-world sample of over 700,000 subjects (of which >230,000 were included in this study). In addition, a large number of variables were considered from biological to self-ratings. The statistical approach is clearly outlined and appropriate.

Results: A detailed results section is provided and supported by figures and tables. In additional supplementary files are available providing additional insight.

Discussion: A well written discussion is provided with interpretation of the findings compared with the literature. Some novel theories and aspects are provided. 

The conclusion is adequate and this should eb used to inform the abstract’s conclusion.

We appreciate Reviewer #1 for his/her positive and productive comments to our manuscript. 

1. Abstract: However, the convulsion should be based on study findings rather than proposed future work (as this has not been evaluated).

We have revised conclusion section to be based on study findings as suggested (Page 2, Line 25).

We found that an increase in BMI was associated with a decrease in the number of residual teeth from younger ages independently of smoking status and diabetes in the large scale of Japanese database.

2. Discussion: A well written discussion is provided with interpretation of the findings compared with the literature. Some novel theories and aspects are provided. The conclusion is adequate and this should eb used to inform the abstract’s conclusion.

We have used this conclusion also in the Abstract with minor modification as suggested. 

Reviewer #2: 

Few suggestions to be included

1. In statistical analysis, it is better to include full name of SPSS along with release date and version.

All analyses were performed using SPSS (Statistical Package for Social Science) Statistics Ver. 26 (IBM, Inc., Armonk, NY, USA, Released on April 9, 2019), and P-values <0.05 were considered significant based on two-sided tests. We added these sentences in the Method section (Page 5, Line 109).

2. According to WHO, BMI >30 is considered as obese and BMI >25 and <30 is considered as overweight. In your statistical analysis, Obesity has been considered as BMI ≥25. Please justify.

We thank Reviewer #2 for this important point. We referred to the guidelines for the management of obesity disease 2016 (Japan Society For The Study Of Obesity) because we used a large database of the “Japanese” population. Due to the difference in the physical constitution, the guidelines indicate that regardless of gender, BMI 18.5 or more and less than 25 is defined as “normal weight” and 25.0 or more is defined as “obesity”. We added these sentences in the method section (Page 4, Line 86).

---

## [Decision Letter · Decision Letter 1]

30 Aug 2022

Real-world evidence of the impact of obesity on residual teeth in the Japanese population: A cross-sectional study

PONE-D-21-34039R1

Dear Dr. Morino,

We’re pleased to inform you that your manuscript has been judged scientifically suitable for publication and will be formally accepted for publication once it meets all outstanding technical requirements.

Kind regards,

Yann Benetreau, PhD

Division Editor (Staff Editor)

PLOS ONE

Additional Editor Comments (optional):

Reviewers' comments:

Reviewer's Responses to Questions

**Comments to the Author**

1. If the authors have adequately addressed your comments raised in a previous round of review and you feel that this manuscript is now acceptable for publication, you may indicate that here to bypass the “Comments to the Author” section, enter your conflict of interest statement in the “Confidential to Editor” section, and submit your "Accept" recommendation.

Reviewer #2: All comments have been addressed

2. Is the manuscript technically sound, and do the data support the conclusions?

Reviewer #2: Yes

3. Has the statistical analysis been performed appropriately and rigorously? 

Reviewer #2: Yes

4. Have the authors made all data underlying the findings in their manuscript fully available?

Reviewer #2: Yes

5. Is the manuscript presented in an intelligible fashion and written in standard English?

Reviewer #2: Yes

6. Review Comments to the Author

Reviewer #2: The authors have responded to the comments and given justification with appropriate reference. The manuscript is accepted for publication.

7. PLOS authors have the option to publish the peer review history of their article (what does this mean?). If published, this will include your full peer review and any attached files.

Reviewer #2: **Yes: **Shilpa Gunjal

---

## [Editor Report · Acceptance letter]

1 Sep 2022

PONE-D-21-34039R1 

Real-world evidence of the impact of obesity on residual teeth in the Japanese population: A cross-sectional study 

Dear Dr. Morino:

I'm pleased to inform you that your manuscript has been deemed suitable for publication in PLOS ONE. Congratulations! Your manuscript is now with our production department. 

Kind regards, 

on behalf of

Dr. Yann Benetreau 

Staff Editor

PLOS ONE